# Changes in Invasive Pneumococcal Disease Caused by *Streptococcus pneumoniae* Serotype 1 following Introduction of PCV10 and PCV13: Findings from the PSERENADE Project

**DOI:** 10.3390/microorganisms9040696

**Published:** 2021-03-27

**Authors:** Julia C. Bennett, Marissa K. Hetrich, Maria Garcia Quesada, Jenna N. Sinkevitch, Maria Deloria Knoll, Daniel R. Feikin, Scott L. Zeger, Eunice W. Kagucia, Adam L. Cohen, Krow Ampofo, Maria-Cristina C. Brandileone, Dana Bruden, Romina Camilli, Jesús Castilla, Guanhao Chan, Heather Cook, Jennifer E. Cornick, Ron Dagan, Tine Dalby, Kostas Danis, Sara de Miguel, Philippe De Wals, Stefanie Desmet, Theano Georgakopoulou, Charlotte Gilkison, Marta Grgic-Vitek, Laura L. Hammitt, Markus Hilty, Pak-Leung Ho, Sanjay Jayasinghe, James D. Kellner, Jackie Kleynhans, Mirjam J. Knol, Jana Kozakova, Karl G. Kristinsson, Shamez N. Ladhani, Laura MacDonald, Grant A. Mackenzie, Lucia Mad’arová, Allison McGeer, Jolita Mereckiene, Eva Morfeldt, Tuya Mungun, Carmen Muñoz-Almagro, J. Pekka Nuorti, Metka Paragi, Tamara Pilishvili, Rodrigo Puentes, Samir K. Saha, Aalisha Sahu Khan, Larisa Savrasova, J. Anthony Scott, Anna Skoczyńska, Shigeru Suga, Mark van der Linden, Jennifer R. Verani, Anne von Gottberg, Brita A. Winje, Inci Yildirim, Khalid Zerouali, Kyla Hayford

**Affiliations:** 1Johns Hopkins Bloomberg School of Public Health, Baltimore, MD 21205, USA; mhetric2@jhmi.edu (M.K.H.); mgarci64@jhmi.edu (M.G.Q.); jsinkev1@jhu.edu (J.N.S.); sz@jhu.edu (S.L.Z.); lhammitt@jhu.edu (L.L.H.); kylahayford@jhu.edu (K.H.); 2Independent Consultant, 1296 Coppet, Switzerland; drf3217@gmail.com; 3KEMRI-Wellcome Trust Research Programme, Epidemiology and Demography Department, Centre for Geographic Medicine-Coast, P.O. Box 230-80108 Kilifi, Kenya; EKagucia@kemri-wellcome.org (E.W.K.); anthony.scott@lshtm.ac.uk (J.A.S.); 4World Health Organization, 1202 Geneva, Switzerland; dvj1@cdc.gov; 5Division of Pediatric Infectious Diseases, Department of Pediatrics, University of Utah Health Sciences Center, Salt Lake City, UT 84132, USA; Krow.Ampofo@hsc.utah.edu; 6National Laboratory for Meningitis and Pneumococcal Infections, Center of Bacteriology, Institute Adolfo Lutz (IAL), São Paulo 01246-902, Brazil; maria.brandileone@ial.sp.gov.br; 7Arctic Investigations Program, Division of Preparedness and Emerging Infections, National Center for Emerging and Zoonotic Infectious Diseases, Centers for Disease Control and Prevention, Anchorage, AK 99508, USA; zkg9@cdc.gov; 8Department of Infectious Diseases, Italian National Institute of Health (Istituto Superiore di Sanità, ISS), 00161 Rome, Italy; romina.camilli@iss.it; 9CIBER Epidemiología y Salud Pública (CIBERESP), 28029 Madrid, Spain; jcastilc@navarra.es (J.C.); cma@sjdhospitalbarcelona.org (C.M.-A.); 10Instituto de Salud Pública de Navarra—IdiSNA, 31003 Pamplona, Navarra, Spain; 11Singapore Ministry of Health, Communicable Diseases Division, Singapore 308442, Singapore; CHAN_Guanhao@moh.gov.sg; 12Centre for Disease Control, Department of Health and Community Services, Darwin, NT 8000, Australia; hcdarwin@gmail.com; 13Institute of Infection, Veterinary & Ecological Sciences, University of Liverpool, Liverpool CH64 7TE, UK; jcornick@mlw.mw; 14Malawi-Liverpool-Wellcome Trust Clinical Research Programme, Chichiri, P.O. Box 30096 Blantyre, Malawi; 15Faculty of Health Sciences, Ben-Gurion University of the Negev, 8410501 Beer-Sheva, Israel; rdagan@bgu.ac.il; 16Bacteria, Parasites and Fungi, Statens Serum Institut, DK-2300 Copenhagen, Denmark; TID@ssi.dk; 17Santé Publique France, the French National Public Health Agency, Saint Maurice CEDEX, 94415 Paris, France; Costas.DANIS@santepubliquefrance.fr; 18Epidemiology Department, Dirección General de Salud Pública, 28009 Madrid, Spain; sarade.miguel@salud.madrid.org; 19Department of Social and Preventive Medicine, Laval University, Québec, QC G1V 0A6, Canada; philippe.dewals@criucpq.ulaval.ca; 20Department of Microbiology, Immunology and Transplantation, KU Leuven, BE-3000 Leuven, Belgium; stefanie.desmet@uzleuven.be; 21National Reference Centre for Streptococcus Pneumoniae, University Hospitals Leuven, 3000 Leuven, Belgium; 22National Public Health Organisation, 15123 Athens, Greece; t.georgakopoulou@eody.gov.gr; 23Epidemiology Team, Institute of Environmental Science and Research, Porirua, Wellington 5240, New Zealand; Charlotte.Gilkison@esr.cri.nz; 24Communicable Diseases Centre, National Institute of Public Health, 1000 Ljubljana, Slovenia; Marta.Vitek@nijz.si; 25Swiss National Reference Centre for Invasive Pneumococci, Institute for Infectious Diseases, University of Bern, 3012 Bern, Switzerland; Markus.Hilty@ifik.unibe.ch; 26Department of Microbiology and Carol Yu Centre for Infection, The University of Hong Kong, Hong Kong, China; plho@hku.hk; 27National Centre for Immunisation Research and Surveillance and Discipline of Child and Adolescent Health, Children’s Hospital Westmead Clinical School, Faculty of Medicine and Health, University of Sydney, Westmead, NSW 2145, Australia; sanjay.jayasinghe@health.nsw.gov.au; 28Department of Pediatrics, University of Calgary, and Alberta Health Services, Calgary, AB T3B 6A8, Canada; kellner@ucalgary.ca; 29Centre for Respiratory Diseases and Meningitis, National Institute for Communicable Diseases of the National Health Laboratory Service, Johannesburg 2192, South Africa; JackieL@nicd.ac.za (J.K.); annev@nicd.ac.za (A.v.G.); 30School of Public Health, Faculty of Health Sciences, University of the Witwatersrand, Johannesburg 2000, South Africa; 31National Institute for Public Health and the Environment, 3721 MA Bilthoven, The Netherlands; mirjam.knol@rivm.nl; 32National Institute of Public Health (NIPH), 100 42 Praha, Czech Republic; jana.kozakova@szu.cz; 33Department of Clinical Microbiology, Landspitali—The National University Hospital, Hringbraut, 101 Reykjavik, Iceland; karl@landspitali.is; 34Immunisation and Countermeasures Division, Public Health England, London NW9 5EQ, UK; shamez.ladhani@phe.gov.uk; 35Public Health Scotland, Glasgow G2 6QE, UK; laura.macdonald4@phs.scot; 36Faculty of Infectious and Tropical Diseases, London School of Hygiene & Tropical Medicine, Keppel St, London WC1E 7HT, UK; gmackenzie@mrc.gm; 37Medical Research Council Unit the Gambia at London School of Hygiene & Tropical Medicine, P.O. Box 273 Banjul, The Gambia; 38New Vaccines Group, Murdoch Children’s Research Institute, Parkville, Melbourne, VIC 3052, Australia; 39National Reference Centre for Pneumococcal and Haemophilus Diseases, Regional Authority of Public Health, 975 56 Banská Bystrica, Slovakia; madarova@vzbb.sk; 40Toronto Invasive Bacterial Diseases Network, Laboratory Medicine and Pathobiology, University of Toronto, Toronto, ON M5S 1A8, Canada; Allison.McGeer@sinaihealthsystem.ca; 41HSE Health Protection Surveillance Centre, Mountjoy, Dublin D01 A4A3, Ireland; jolita.mereckiene@hse.ie; 42Department of Microbiology, Public Health Agency of Sweden, 171 82 Solna, Sweden; eva.morfeldt@folkhalsomyndigheten.se; 43National Center of Communicable Diseases (NCCD), Ministry of Health, Bayanzurkh District, Ulaanbaatar 13336, Mongolia; tuya_mungun@yahoo.com; 44Medicine Department, Universitat Internacional de Catalunya, 08017 Barcelona, Spain; 45Molecular Microbiology Department, Hospital Sant Joan de Déu Research Institute, 08950 Esplugues de Llobregat, Barcelona, Spain; 46Department of Health Security, Finnish Institute for Health and Welfare, 00271 Helsinki, Finland; pekka.nuorti@tuni.fi; 47Health Sciences Unit, Faculty of Social Sciences, University of Tampere, 33100 Tampere, Finland; 48Centre for Medical Microbiology, National Laboratory of Health, Environment and Food, 2000 Maribor, Slovenia; metka.paragi@nlzoh.si; 49National Center for Immunizations and Respiratory Diseases, Centers for Disease Control and Prevention, Atlanta, GA 30333, USA; tpilishvili@cdc.gov (T.P.); qzr7@cdc.gov (J.R.V.); 50Instituto de Salud Pública de Chile, Santiago 7780050, Santiago Metropolitan, Chile; rpuentes@ispch.cl; 51Child Health Research Foundation, Dhaka 1207, Bangladesh; samirk.sks@gmail.com; 52Ministry of Health and Medical Services, Suva, Fiji; aalisha@gmail.com; 53Centre for Disease Prevention and Control of Latvia, 1005 Riga, Latvia; larisa.savrasova@spkc.gov.lv; 54Doctoral Studies Department, Riga Stradinš University, 1007 Riga, Latvia; 55National Reference Centre for Bacterial Meningitis, National Medicines Institute, 00-725 Warsaw, Poland; a.skoczynska@nil.gov.pl; 56Infectious Disease Center and Department of Clinical Research, National Hospital Organization Mie Hospital, Tsu, Mie 514-0125, Japan; suga.shigeru.ke@mail.hosp.go.jp; 57National Reference Center for Streptococci, Department of Medical Microbiology, University Hospital RWTH Aachen, 52074 Aachen, Germany; mlinden@ukaachen.de; 58Centers for Disease Control and Prevention (CDC), Center for Global Health (CGH), Division of Global Health Protection (DGHP), P.O. Box 606-00621 Nairobi, Kenya; 59School of Pathology, Faculty of Health Sciences, University of the Witwatersrand, Braamfontein, Johannesburg 2000, South Africa; 60Department of Infection Control and Vaccine, Norwegian Institute of Public Health, 0456 Oslo, Norway; Brita.Askeland.Winje@fhi.no; 61Department of Pediatrics, Yale New Haven Children’s Hospital, New Haven, CT 06504, USA; inci.yildirim@yale.edu; 62Bacteriology-Virology and Hospital Hygiene Laboratory, Ibn Rochd University Hospital Centre, Casablanca 20250, Morocco; khalid.zerouali2000@gmail.com; 63Department of Microbiology, Faculty of Medicine and Pharmacy, Hassan II University of Casablanca, Casablanca 20000, Morocco

**Keywords:** invasive pneumococcal disease, pneumococcal conjugate vaccines, serotypes, vaccine impact

## Abstract

*Streptococcus pneumoniae* serotype 1 (ST1) was an important cause of invasive pneumococcal disease (IPD) globally before the introduction of pneumococcal conjugate vaccines (PCVs) containing ST1 antigen. The Pneumococcal Serotype Replacement and Distribution Estimation (PSERENADE) project gathered ST1 IPD surveillance data from sites globally and aimed to estimate PCV10/13 impact on ST1 IPD incidence. We estimated ST1 IPD incidence rate ratios (IRRs) comparing the pre-PCV10/13 period to each post-PCV10/13 year by site using a Bayesian multi-level, mixed-effects Poisson regression and all-site IRRs using a linear mixed-effects regression (N = 45 sites). Following PCV10/13 introduction, the incidence rate (IR) of ST1 IPD declined among all ages. After six years of PCV10/13 use, the all-site IRR was 0.05 (95% credibility interval 0.04–0.06) for all ages, 0.05 (0.04–0.05) for <5 years of age, 0.08 (0.06–0.09) for 5–17 years, 0.06 (0.05–0.08) for 18–49 years, 0.06 (0.05–0.07) for 50–64 years, and 0.05 (0.04–0.06) for ≥65 years. PCV10/13 use in infant immunization programs was followed by a 95% reduction in ST1 IPD in all ages after approximately 6 years. Limited data availability from the highest ST1 disease burden countries using a 3 + 0 schedule constrains generalizability and data from these settings are needed.

## 1. Introduction

*Streptococcus pneumoniae* is a major cause of pneumonia, meningitis, and pleural effusion in children and adults [1,2,3,4]. There are at least 100 known serotypes of pneumococci [5]. Before the introduction of pneumococcal conjugate vaccines (PCVs), serotype 1 (ST1) was one the most common causes of invasive pneumococcal disease (IPD), especially in Asia and Africa, and globally was responsible for approximately 9% of IPD among children <5 years of age [6]. ST1 is distinct from other serotypes in that it has a high invasiveness potential, is not commonly carried in the nasopharynx [7,8], and in some settings occurs in a cyclical pattern, approximately every 3–9 years [9,10,11]. Additionally, ST1 can cause large pneumococcal outbreaks among all ages, including older children and young adults, in the African meningitis belt and other outbreak-prone settings with up to 10–30-fold increases in ST1 cases compared to pre-outbreak baselines [12,13,14,15].

The first PCV licensed for use in infants, seven-valent PCV (Prevenar/Prevnar, Pfizer), did not include ST1 antigen. Since then, the introduction of PCVs containing ST1 antigen (PCV10 [Synflorix, GlaxoSmithKline], PCV13 [Prevenar13/Prevnar13, Pfizer]) into many national infant immunization programs since 2009 has been shown to substantially reduce ST1 IPD and end pneumococcal outbreaks caused by ST1. These effects have been demonstrated among directly immunized children and also unvaccinated older children and adults, through indirect effects, in both high and low IPD burden settings [9,10,12,16,17,18,19,20]. However, in some PCV10/13 using settings ST1 outbreaks continued to occur or ST1 IPD incidence rates did not substantially decline in the early years immediately following PCV10/13 introduction [21,22,23,24]. 

Evaluating the impact of PCV10/13 vaccination on ST1 IPD is challenging in a single surveillance site. In many settings, annual ST1 incidence rates are unstable because case counts are small, particularly after vaccine introduction. Many sites are also limited by short pre- and post-vaccine introduction surveillance periods, further limiting inferences that can be drawn from a single site. Assessing vaccine impact is also confounded by the cyclic nature of ST1 in which it is common to observe multiple years of zero ST1 cases prior to vaccine use. Quantifying the impact of PCV10/13 on ST1, which has several unique characteristics compared to other vaccine-type serotypes included in currently licensed PCVs, is important for policymakers seeking to reduce the burden of ST1 IPD through immunization. The Pneumococcal Serotype Replacement and Distribution Estimation (PSERENADE) project evaluated all available published and unpublished serotype-specific IPD data to estimate the impact of PCV10/PCV13 on ST1 IPD incidence at the global scale.

## 2. Materials and Methods

### 2.1. Data Collection and Eligibility Criteria

IPD surveillance sites with eligible data contributed annual serotype-specific IPD case data and population denominators to the project. A systematic approach to identify eligible sites and request data is described in detail elsewhere [25]. ST1 IPD was defined as the isolation of *Streptococcus pneumoniae* from a normally sterile site or detection of pneumococcus in cerebrospinal fluid (CSF) or pleural fluid using *lytA*-based polymerase chain reaction (PCR), or antigen testing confirmed as ST1. Sites with ST1 IPD case counts and population denominators that met eligibility criteria were included in the analysis (Box 1, Table 1, Appendix A).

Box 1Inclusion criteria.
Site reports annual ST1 IPD incidence data:
-ST1 case counts by age group, and-Population-based denominators by age group.
At least 50% of isolates serotyped for included years by age group.At least one complete year of data post-PCV10/13 introduction, excluding the year of introduction.At least 50% uptake for primary PCV series at 12 months of age in at least one year post-PCV10/13 introduction.PCV10 or PCV13 is universally recommended for all infants in the national infant immunization schedule.No major changes or biases in surveillance that would affect estimates of ST1 incidence rates.


Two PSERENADE coordinators conducted a standard data quality review for each site to evaluate if surveillance system changes or other factors besides PCV introductions influenced incidence rates (IR) of IPD over available years of surveillance data [25]. After review and discussion with site investigators, certain site-year-age group data were excluded if determined to fall within periods of differential surveillance capture or if the impact of changes in surveillance protocols on IPD IRs could not be accounted for in the analysis. For all sites, we defined the year of PCV introduction as the year PCV10/13 was universally introduced if PCV was introduced in the first three quarters of the year, or as the following calendar year if otherwise. For data submitted in epidemiologic years rather than calendar years, the introduction year was defined accordingly. For all sites, the year of PCV10/13 introduction was defined as ‘year 0’ for the analyses.

### 2.2. Data Analysis

#### 2.2.1. Adjustments for Missing Data

Adjustments for missing serotype data assume that missing serotype data are missing completely at random, that is the serotype distribution of serotyped cases is not biased or different from the serotype distribution of cases that were not serotyped or not fully serotyped. Site-year-age group strata that violated this assumption or reported serotypes for less than 50% of cases were excluded from the ST1 analysis for that stratum. For cases that were reported as not serotyped (serotyping was not attempted for any reason), the population denominators were adjusted by the proportion of cases that were serotyped (i.e., annual denominator * percent of cases that were serotyped in that year) for each site by year and age group. Because the proportion of cases serotyped varies across sites, population denominators were adjusted rather than reapportioning serotypes to unknown serotype cases in order to give appropriate weight to sites in the model based on serotype data reported. If ST1 and a second serotype was reported for a case, it was included as an ST1 case. Cases reported as a serotype pool which includes ST1 (e.g., pool A) were excluded. For cases with unknown age, the population denominators were adjusted by the proportion of cases with known age (i.e., annual denominator * percent of cases with known age in that year) for each year and age group. Minor changes were made to the cut-offs for age groups when standard age categories used for analyses were not available from the site.

#### 2.2.2. Statistical Analysis

Annual ST1 IPD incidence rate ratios (IRRs) comparing the pre-PCV10/13 period to each post-PCV10/13 year were estimated by age group and for all ages in a three-step process. First, ST1 IR curves were estimated over years of available data for each site using a Bayesian multi-level, mixed-effects Poisson regression using the MCMCglmm package in R [26]. The model included data from all sites (using either PCV10 or PCV13) with an offset for population denominator and random effects for all of the site-specific regression coefficients, which allows for heterogeneity among sites in the shapes of their incidence curves. Sites using PCV10 and PCV13 were modeled together to increase sample size and as no difference in impact on ST1 IPD was observed by product (Appendix A). The regression identified commonalities within and across sites in the direction of change over time and smoothed out observed annual variability. Data points from the same site were treated as repeated measures over time and sites with small case counts or few years of data had less influence than sites with larger case counts and many years of data.

ST1 outbreaks tended to occur in a cyclical pattern prior to the introduction of PCV10/13. The model did not account for outbreaks occurring in a cyclical pattern. Therefore, in order to generate an expected baseline ST1 IPD IR in any given year, the regression modeled pre-PCV10/13 IRs as a single mean rate with a slope of zero to capture an ‘average’ pre-PCV10/13 ST1 IR. PCV7 years of use were included in the pre-PCV10/13 period as no consistent impact of PCV7 on ST1 IRs, either increases (i.e., serotype replacement) or decreases, were observed across sites, as expected given pre-PCV10/13 ST1 carriage patterns [7]. This increased the number of pre-PCV10/13 years included in the analysis and better captured the baseline ST1 IR. For each site, a non-linear break (allowing an abrupt hinge in the curve) was included in the model one year prior to PCV10/13 introduction to capture the change from the pre-PCV10/13 period to the year of PCV10/13 introduction and cubic splines knots (allowing a smooth change in the slope) were included for each site at years +1 and +3 (the second and fourth year of PCV10/13 use) to allow for flexibility in the IR of ST1 over time for each site following PCV10/13 introduction. Site-specific modeled ST1 IR curves were visually inspected for model fit and approved by site investigators with expertise in IPD surveillance at each site.

Second, the pre-PCV10/13 ST1 IR was used as a counterfactual ST1 IR (i.e., an expected ST1 IR in any given post-PCV10/13 year in the absence of PCV10/13 introduction) for sites with both pre- and post-PCV10/13 data. The site-specific modeled ST1 IR and counterfactual IR were used to estimate site-specific annual IRRs in each post-PCV10/13 year (reported as the mean of the posterior distribution of rate ratios) for each site. Site-specific IRRs were not generated for sites without pre-PCV10/13 years of data. Credibility intervals (CIs, Bayesian confidence interval analog) were estimated using the 2.5 and 97.5 percentiles of the posterior distribution of the IRs (Appendix A).

Finally, modeled site-specific IRRs were used to estimate all-site weighted average IRRs in each post-PCV10/13 year using a linear mixed-effects regression where site-specific IRRs were regressed on time since PCV10/13 introduction and weighted to give more influence to sites whose IRR standard errors were smaller. In sensitivity analyses, the all-site weighted average IRRs were estimated restricting to sites with data in all age groups and after adjusting the counterfactual IR by all-serotype IPD pre-PCV trends. All analyses were conducted in R (R Core Team, 2019).

## 3. Results

### 3.1. Description of Sites and Included Data

Of the 52 sites that met data collection eligibility criteria and contributed data to the PSERENADE project, 45 were included in the serotype 1 analysis (41 for children <5 years of age, 38 for 5–17 years of age, 37 for 18–49 years of age, 36 for 50–64 years of age, and 36 for ≥65 years of age). Two sites were excluded due to their population-based surveillance being restricted to pneumococcal meningitis, four sites were excluded due to a combination of biases in the surveillance system over time, such as changed to surveillance protocols, that could not be accounted for in the analysis and/or less than 50% of cases being serotyped, and one site was excluded due to zero ST1 cases being reported in all years of available data. Additionally, several age groups from included sites did not meet eligibility criteria and were excluded (Appendix A).

Seven sites (16%) included in the analysis used PCV10, 24 (53%) used PCV13, and 14 (31%) used a combination of PCV10 and PCV13 in the infant PCV program. Only 14 (31%) sites introduced PCV10 or PCV13 into the routine immunization schedule with a catch-up campaign. The majority of sites used a PCV schedule including a booster dose (40, 89% used a 2 + 1 or 3 + 1 schedule and 5, 11% used a 3 + 0 schedule). Nearly half were from Europe (22 (49%)), 8 (18%) were from North America, 5 (11%) from Sub-Saharan Africa, 3 (7%) from Oceania, 3 (7%) from Asia, 2 (4%) from Latin America and the Caribbean and 2 (4%) from Northern Africa and Western Asia. The median PCV10/13 uptake for all years of available data after PCV10/13 introduction was 92% (range: 55–98%) (Table 1).

Of included sites with available data on specimen type, the median proportion of all ST1 IPD cases from CSF was 1.4% (range: 0–55.5%). Annual site-specific ST1 IRRs were estimated for 40 (89%) sites with both pre- and post-PCV10/13 ST1 surveillance data. The median number of surveillance years included in the analysis was 7 (range: 0–19) prior to the introduction of PCV10/13 and 8 (range: 2–10) after the introduction of PCV10/13 (including the year of PCV10/13 introduction). The median proportion of cases serotyped annually was 94% (range: 50–100%). The median number of ST1 cases included in the analysis per site was 29 (range: 1–499) for children <5 years of age, 46 (range: 2–768) for 5–17 years of age, 51 (range: 1–1776) for 18–49 years of age, 25 (range: 1–753) for 50–64 years of age, and 26 (range: 1–748) for ≥65 years of age (Table 1, Figure 1).

### 3.2. Impact of PCV10/13 on ST1 Incidence

All-site weighted average ST1 IPD IRRs comparing the pre-PCV10/13 period to each post-PCV10/13 year are shown in Table 2 and Figure 2. The all-site weighted average IRRs in the year of PCV10/13 introduction by age group ranged from 0.82 to 1.09 and was 1.09 (95% CI: 0.92–1.29) for children <5 years of age, 1.06 (0.88–1.28) for 5–17 years of age, 0.94 (0.73–1.22) for 18–49 years of age, 0.85 (0.70–1.04) for 50–64 years of age, and 0.82 (0.68–0.99) for ≥65 years of age. The ST1 IRR declined for every age group in each subsequent post-PCV10/13 year. By the sixth year of PCV10/13 use (year +5 post-PCV10/13 introduction), the all-site weighted average IRR compared to the pre-PCV10/13 period was 0.05 (0.04–0.06) for all ages, or a 95% relative reduction in ST1 IPD compared to the pre-PCV10/13 period. The reduction in ST1 IPD for each age group ranged from 92% to 95% in the sixth year of PCV10/13 use: IRR 0.05 (0.04–0.05) for children <5 years of age, 0.08 (0.06–0.09) for 5–17 years of age, 0.06 (0.05–0.08) for 18–49 years of age, 0.06 (0.05–0.07) for 50–64 years of age, and 0.05 (0.04–0.06) for ≥65 years of age.

In the early years of PCV10/13 use, site-specific IRRs were heterogeneous. Some sites reported outbreaks or had elevated levels of ST1 IPD around the time of PCV10/13 introduction, including two sites with very small sample sizes and large proportion increases in ST1 IRs. Other sites had little to no ST1 disease at the time of PCV10/13 introduction compared to the pre-PCV10/13 ST1 IRs. After five years of PCV10/13 use (year +4 post-PCV10/13), the impact of PCV10/13 on ST1 IPD was homogeneous across all included sites and age groups. No ST1 outbreaks were observed after five or more years of PCV10/13 use in any site (Figure 3). Results were similar when analyses were restricted to sites with data in all age groups (results not shown), when sites with very small sample size were excluded (results not shown), and after adjusting the counterfactual IR by all-serotype IPD pre-PCV trends (Appendix A). No differences in ST1 impact were observed by visual inspection among the included sites by PCV product, region, infant PCV schedule, or adult pneumococcal polysaccharide vaccine recommendation (Appendix A). One site, which was excluded from the analytic model because the dataset was limited to meningitis cases, observed declines in ST1 pneumococcal meningitis IRs after PCV10 introduction that were consistent with declines seen in ST1 IPD in the other sites (Appendix A).

## 4. Discussion

Our analysis demonstrates that there have been large and sustained decreases in ST1 IPD among both children targeted for immunization and among unvaccinated older children and adults through indirect effects. We used a standardized approach to analyze data from 45 surveillance sites and analytic methods that strengthened predictions from sites with few years of data and small sample sizes by borrowing strength from the overall trends observed across all sites. This allowed sites with few years of data and small sample sizes to still contribute proportionately to the analysis where data were available. As a result, this analysis is the most comprehensive assessment of changes in ST1 IPD after PCV10/13 introduction and demonstrates homogeneity in long-term impact of PCV10/13 on ST1 IPD across sites. These results were used to inform global vaccine policy recommendations around the use of pneumococcal vaccines in community outbreak settings [27].

The all-site weighted average IRRs are consistent with findings from individual surveillance sites on the long-term impact of PCV10/13 on ST1 IPD [9,10,12,16,17,18,19,20]. In the first several years of PCV10/13 use, the observed impact of PCV10/13 on ST1 IPD was heterogeneous, in part, due to the cyclic and outbreak nature of ST1 IPD and likely reflects heterogeneity in pre-PCV10/13 temporal trends with respect to the timing of PCV10/13 introduction. In some sites, ST1 IPD rates in the early years were greater than the pre-PCV10/13 average (because cyclical increases or outbreaks occurred at the time of or immediately following PCV10/13 introduction or because of noise in small datasets) and in other sites ST1 IPD rates were lower than the pre-PCV10/13 average immediately following PCV10/13 introduction. However, further into the PCV10/13 period, every site had sustained reductions in ST1 IPD below the pre-PCV10/13 rate. Prior to PCV10/13 introduction ST1 was known to cause severe disease to a greater degree in older children and younger adults compared to other serotypes [3,13,28] and importantly, we observed substantial reductions in ST1 IPD for all age groups. There was concern prior to the widespread introduction of PCV10/13 regarding the immunogenicity of PCV10/13 when used without a booster dose against ST1 [29]. Although only five sites using a 3 + 0 schedule were included in the analysis, the direct and indirect effects for ST1 IPD after several years of PCV10/13 use in these sites were consistent with patterns observed in sites using a booster dose schedule.

Although not observed in all sites and CIs overlap, our results showed slightly smaller declines in ST1 IPD for children <18 years compared to adults ≥18 years in the year of PCV10/13 introduction, which is contradictory to expected patterns of indirect effects among non-immunized adults following introduction of an infant vaccine [30]. This may reflect secular trends unrelated to vaccine introduction or differences in the hospital and surveillance systems between adults and pediatrics and an increased focus on pediatric surveillance around the time of pediatric vaccine introduction leading to greater detection of pediatric cases compared to adults. Ninety-two percent of sites with adult ST1 data included in the analysis have an adult pneumococcal polysaccharide vaccine recommendation. Although this may have reduced the burden of ST1 IPD among vaccinated adults prior to infant PCV10/13 programs, this does not explain observed patterns in the year of PCV10/13 introduction. The majority of adult polysaccharide vaccine programs began many years prior to the introduction of PCV10/13, recommendations vary by site for adult pneumococcal vaccine use, and data on vaccine uptake among adults was limited. We were not able to detect differences by adult pneumococcal vaccine program recommendation. Despite this, we see substantial and sustained declines in ST1 IPD for all age groups in the following years of PCV10/13 use.

To understand the impact of PCV10/13 introduction, data were restricted to sites with at least 50% uptake for the primary PCV series at 12 months of age in at least one-year post-PCV10/13 introduction and majority of included sites had high PCV uptake. This resulted in most data coming from high-income countries and limited inferences can be made to other regions or areas with lower vaccine uptake. Further, the majority of the data are from sites that used a booster dose. Among the five sites with a 3 + 0 schedule, four introduced PCV10/13 with a catch-up program. Therefore, any added effects of a booster dose and catch-up programs could not be assessed, and results may not be reflective of other settings. In particular, data were limited from areas prone to pneumococcal meningitis outbreaks, such as the African meningitis belt. Only one site from the African meningitis belt, The Gambia, was included in the analysis where a 3 + 0 schedule of PCV13 was introduced without a catch-up program. Although there were few ST1 cases (*n* = 71), ST1 trends for children <18 years of age were consistent with other non-meningitis belt countries in Africa and other regions. In the 4 other sites that used a 3 + 0 schedule (all of which introduced PCV10/13 with a catch-up campaign), ST1 trends were also similar to those observed in sites using a 2 + 1 or 3 + 1 schedule among both children and adults. Two meningitis belt countries with documented pneumococcal outbreaks after PCV13 introduction with a 3 + 0 schedule, Ghana and Burkina Faso, did not contribute data to the PSERENADE project. As in The Gambia, the proportion of ST1 cases occurring among children <5 years of age decreased compared to the pre-PCV13 period in Ghana and Burkina Faso [22,23,24]. However, pneumococcal meningitis outbreaks in persons >5 years of age were documented four years after PCV13 introduction in the Brong-Ahafo region of Ghana (outside of the traditional meningitis belt) [22] and five years after introduction in the Upper West and Northern regions of Ghana (within the traditional meningitis belt) [23]. In both of these outbreaks a large proportion of cases were due to ST1 (between 62–80%) [22,23]. PCV13 uptake in these specific communities was undocumented and national PCV13 uptake in the first two years of use was low in Ghana (41–68%) [22]. In Burkina Faso after 3 years of PCV13 use, ST1 meningitis rates declined by 59% for children <1 year of age, by 25% for children 1–4 years of age, and by 8–17% for individuals ≥5 years of age. Slightly larger declines were observed for all PCV13 serotype meningitis (76% decline for children <1 year, 58% decline for children 1–4 years, and 14–20% decline for individuals ≥5 years of age) [24]. The remaining PCV13 serotype meningitis among individuals ≥1 year of age indicates that indirect effects have not been fully achieved for all vaccine serotypes, including but not limited to ST1, and the 59% decline in ST1 disease among children <1 year of age suggests that after 3 years of use the PCV program has not yet sufficiently protected children targeted for immunization. Although the association between PCV uptake and indirect effects are not well understood, this may indicate low vaccine uptake. The persistence of ST1 IPD in unvaccinated persons in the first five years of PCV10/13 use is consistent with our results, as ST1 outbreaks were still observed in some sites during the first five years of PCV10/13 use and significant declines in ST1 IPD were not observed for some sites until after 5 years of PCV10/13 use (Figure 3). As recommended by WHO, continuation of comprehensive, high-quality serotype-specific IPD surveillance and vaccine uptake monitoring in the African meningitis belt sites still experiencing ST1 outbreaks in the post-PCV period and in countries with suboptimal PCV10/13 uptake could improve understanding of ST1 in these settings with schedules lacking a booster dose or with low PCV10/13 uptake [31].

This analysis was also limited in its ability to model the counterfactual ST1 IR in the absence of PCV10/13. An ideal ST1 counterfactual IR would have modeled the cyclical pattern of ST1 IPD in the absence of PCV10/13 introduction as a baseline comparison for each post-PCV10/13 year, as has been done for single site analyses, but is challenging without monthly data [11]. Due to the number of available years of pre-PCV data and small ST1 sample size, this was not possible for the majority of sites and instead an average pre-PCV10/13 ST1 IR was used as the counterfactual ST1 IR. Using the average pre-PCV10/13 ST1 IR would most likely lead to less valid effect estimates in the early years of PCV10/13 use and may contribute to unexplained differences in IRRs between age groups in the year of PCV10/13 introduction. However, this would have limited impact on the estimates in later post-PCV10/13 later years. Although a high proportion of the cases from included sites were fully serotyped, another limitation of this analysis, which cannot be tested, is the assumption that the prevalence of ST1 among cases that were serotyped is not biased from the prevalence of ST1 cases among cases that were not serotyped or not fully serotyped. Finally, the number of sites with post-PCV10/13 data declined over time and sites with longer follow-up periods tend to be from high-income countries that generally introduced PCV10/13 earlier than low- and middle-income countries. Eleven sites had data through the ninth year of PCV10/13 use and only three sites had data in the tenth year of PCV10/13 use.

These results can provide important context for evaluating the impact of PCV10/13 on other individual serotypes. ST1 is unique from other vaccine-serotypes in its invasiveness potential, carriage patterns, ability to cause large outbreaks among all ages, and association with meningitis [7,8,12,13,14,15]. Future analyses using the PSERENADE dataset will evaluate the impact of PCV10/13 on other individual vaccine and non-vaccine serotypes.

## 5. Conclusions

The introduction of PCV10/13 into infant immunization programs has been associated with the near elimination of ST1 IPD in all ages after approximately 6 years of use, including in settings without a booster dose schedule but with high PCV10/13 uptake, where data are available. Improved population-level serotype-specific IPD surveillance for all ages, including for meningitis, is needed from settings using a 3 + 0 schedule with a history of ongoing ST1 outbreaks in the post-PCV10/13 period, particularly the African meningitis belt, and in countries with suboptimal PCV10/13 uptake. This would allow for a more comprehensive evaluation of the indirect effects of PCV10/13 in older children and adults living in high burden settings using a 3 + 0 schedule or with low PCV10/13 uptake.

## Figures and Tables

**Figure 1 microorganisms-09-00696-f001:**
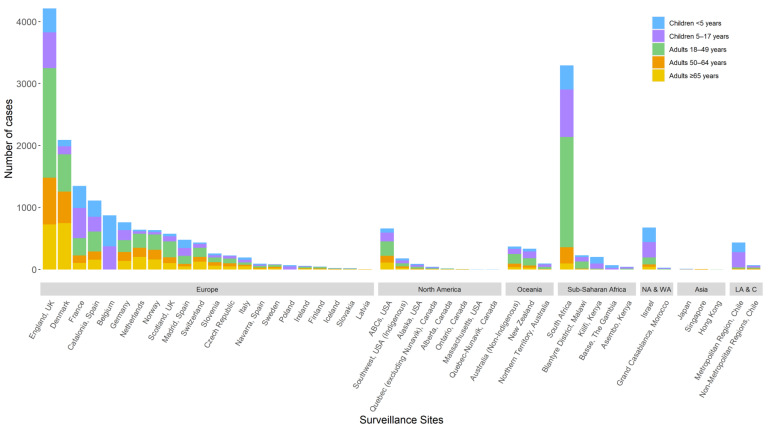
Number of serotype 1 cases per site included in the analysis by region and age group. NA & WA–Northern Africa and Western Asia; LA & C–Latin America and the Caribbean. Not all age groups were included for all sites (Appendix A). Analyses were done with minor changes to age groups for certain sites to align with availability of population denominators and age groups provided by sites in aggregate: the <5 years age group includes 0–5 years from Morocco; the 5–17 years age group included 5–14 years from Japan and Kilifi, Kenya, 5–15 years from Germany, 6–14 years from Morocco, and 5–19 years from Australia and Malawi; and the 18–49 years age group includes 15–49 years from Japan and Kilifi, Kenya, 15–59 years from Morocco, 16–49 years from Germany, and 20–49 years from Australia and Malawi.

**Figure 2 microorganisms-09-00696-f002:**
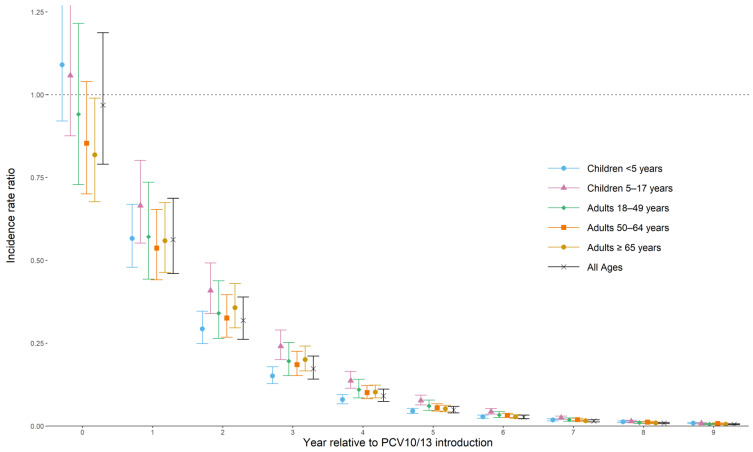
All-site weighted average incidence rate ratios for serotype 1 invasive pneumococcal disease for all ages and by age group. All ages’ analysis (in black) is not an average of each age-specific estimate in each year but rather a re-analysis of the total cases from all ages reporting at each site.

**Figure 3 microorganisms-09-00696-f003:**
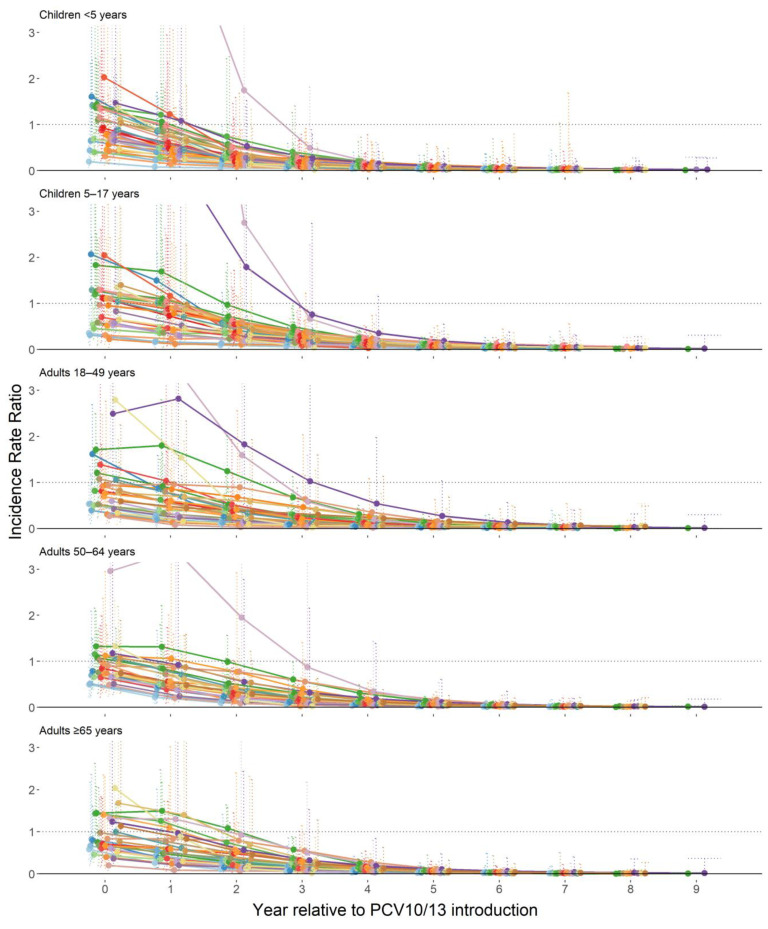
Site-specific modeled serotype 1 invasive pneumococcal disease incidence rate ratios comparing each post-PCV10/13 year to pre-PCV10/13 average, by age group.

**Table 1 microorganisms-09-00696-t001:** Description of infant pneumococcal conjugate vaccine program and surveillance data for included sites. Ordered by vaccine product and schedule.

						Mean PCV10/13 Uptake (%)		Included in ST1 Analysis			Surveillance Years Pre- and Post-PCV10/13 (n)	Proportion ST1 IPD Cases from CSF (%)
Site	PCV10 Period	PCV13 Period	PCV10/13 Schedule	PCV7 Use	PCV10/13 Catch-Up	Primary Series *	WUENIC PCV3 ***	0–17 Years	≥18 Years	ST1 Cases Included in Analysis (n)		
**Finland**	2010–	--	2 + 1	N	N	95	90	Y	Y	46	Pre: 6 Post: 8	4.3
**Iceland**	2011–	--	2 + 1	N	N	97	89	Y	Y	22	Pre: 16 Post: 8	0.0
**Latvia**	2012–	--	2 + 1	Y	N	91	83	N ^b^	Y; ≥50y	5	Pre: 0 Post: 7	20.0
**Slovenia**	2015–2019	2019–	2 + 1	N	N	55	55	Y	Y	259	Pre: 6 Post: 4	0.0
**Netherlands**	2011–	--	3 + 1/2 + 1	Y	N	95	94	Y	Y	642	Pre: 7 Post: 8	2.2
**Asembo, Kenya**	2011–	--	3 + 0	N	Y	86	78	Y	Y; 18–49y	43	Pre: 1 Post: 8	NA
**Kilifi, Kenya**	2011–	--	3 + 0	N	Y	82	78	Y	Y; 18–64y	204	Pre: 11 Post: 6	19.6
**Japan**	--	2013–	3 + 1	Y	N	94 **	98	Y	Y; ≥65y	11	Pre: 4 Post: 5	0.0
**ABCs, USA**	--	2010–	3 + 1	Y	Y	88	93	Y	Y	664	Pre: 12 Post: 8	0.6
**Alaska, USA**	--	2010–	3 + 1	Y	Y	83	93	Y	Y	92	Pre: 19 Post: 8	0.0
**Massachusetts, USA**	--	2010–	3 + 1	Y	Y	94	93	Y; <5y	NA	1	Pre: 8 Post: 8	0.0
**Southwest, USA (Indigenous)**	--	2010–	3 + 1	Y	Y	82	93	Y	Y	180	Pre: 15 Post: 9	2.2
**Alberta, Canada**	--	2010–	2 + 1	Y	N	88 **	77	Y; <5y	Y	16	Pre: 10 Post: 8	0.0
**Denmark**	--	2010–	2 + 1	Y	N	91 **	93	Y	Y	2089	Pre: 10 Post: 9	2.2
**France**	--	2010–	2 + 1	Y	N	93	91	Y	Y	1346	Pre: 9 Post: 9	5.9
**Ireland**	--	2010–	2 + 1	Y	N	91	91	Y	Y	58	Pre: 3 Post: 8	0.0
**Israel**	--	2010–	2 + 1	Y	N	95	93	Y	Y	677	Pre: 8 Post: 8	3.4
**Italy**	--	2010–	2 + 1	Y	N	86 **	87	Y	Y	193	Pre: 0 Post: 9	6.7
**Norway**	--	2011–	2 + 1	Y	N	93	93	Y	Y	637	Pre: 7 Post: 7	1.4
**Singapore**	--	2011–	2 + 1	Y	Y	84	74	N ^d^	Y; ≥50y	8	Pre: 2 Post: 8	0.0
**South Africa**	--	2011–	2 + 1	Y	Y	77 **	77	Y	Y	3292	Pre: 6 Post: 8	38.2
**Madrid, Spain**	--	2010–	2 + 1	Y	N	98	93	Y	Y	479	Pre: 3 Post: 9	0.8
**Switzerland**	--	2010–	2 + 1	Y	Y	79 **	77	Y	Y	436	Pre: 8 Post: 7	0.5
**England, UK**	--	2010–	2 + 1	Y	N	94	92	Y	Y	4214	Pre: 10 Post: 10	1.5
**Scotland, UK**	--	2010–	2 + 1	Y	N	97	92	Y	Y	578	Pre: 10 Post: 9	NA
**Germany**	--	2009–	3 + 1/2 + 1	Y	N	85	93	Y	Y	760	Pre: 5 Post: 9	4.1
**Catalonia, Spain**	--	2010–2015 ^a^ 2016–	3 + 1/2 + 1	Y ^a^	N	70	93	Y	Y	1111	Pre: 4 Post: 8	1.5
**Navarra, Spain**	--	2010–2015 ^a^ 2016–	3 + 1/2 + 1	Y ^a^	N	71	93	Y	Y	93	Pre: 9 Post: 9	0.0
**Australia (Non-Indigenous)**	--	2011–	3 + 0	Y	Y	92	92	Y	Y	371	Pre: 9 Post: 7	0.8
**Basse, The Gambia**	--	2011–	3 + 0	Y	N	77	95	Y	N ^b^	71	Pre: 2 Post: 7	1.4
**Blantyre District, Malawi**	--	2011–	3 + 0	N	Y	92	88	Y	Y	229	Pre: 5 Post: 7	55.5
**Northern Territory, Australia**	2009–2011	2011–	3 + 1	Y	Y	88	92	Y	Y	97	Pre: 16 Post: 8	1.0
**Quebec-Nunavik, Canada**	2009–2010	2011–	3 + 1	Y	N	97	75	Y; <5y	N ^c^	1	Pre: 9 Post: 10	0.0
**Hong Kong**	2010–2011	2011-	3 + 1	Y	N	98	--	N ^d^	Y; 18–49y	1	Pre: 0 Post: 5	0.0
**New Zealand**	2011–2014 2017–	2014–2017	3 + 1	Y	N	93	93	Y	Y	334	Pre: 9 Post: 8	0.6
**Belgium**	2015–2019	2011–2015 2019–	2 + 1	Y	N	95 **	94	Y	NA	872	Pre: 5 Post: 8	1.3
**Poland**	2017–	2017– ^e^	2 + 1	N	N	94	60	Y	N ^b^	69	Pre: 9 Post: 2	4.3
**Quebec (excluding Nunavik), Canada**	2009–2010 2018–	2011–2018	2 + 1	Y	N	97	75	Y	Y	43	Pre: 8 Post: 10	0.0
**Metropolitan Region, Chile**	2011–2015	2016–	2 + 1	Y	N	97	88	Y	Y	437	Pre: 9 Post: 8	2.7
**Non-Metropolitan Regions, Chile**	2011–2017	2017–	2 + 1	N	N	97	89	Y	Y	69	Pre: 0 Post: 7	0.0
**Grand Casablanca, Morocco**	2012–	2010–2012	2 + 1	N	N	91	90	Y	Y; 18–49y	29	Pre: 4 Post: 7	37.9
**Slovakia**	2011–	2011–	2 + 1	Y	Y	97	97	Y	Y	20	Pre: 0 Post: 7	5.0
**Sweden**	2010–	2010–2019	2 + 1	Y	N	97 **	97	Y	Y	84	Pre: 1 Post: 5	NA
**Ontario, Canada**	2009–2010	2010–	3 + 1/2 + 1	Y	Y	72 **	79	N ^d^	Y	9	Pre: 3 Post: 9	0.00
**Czech Republic**	2010–	2010–	3 + 1/2 + 1	N	N	74 **	--	Y	Y	227	Pre: 2 Post: 8	2.2

PCV: Pneumococcal conjugate vaccines. ST1: Serotype 1. CSF: Cerebrospinal fluid. -- Not universally used. Y: Yes; N: No; NA: Not applicable. ^a^ Recommended for high-risk populations only but had substantial (≥50% annually) private market uptake among the general population. ^b^ Biases in surveillance system over time that could not be accounted for. ^c^ Low proportion of cases serotyped. ^d^ Zero ST1 cases in all years. ^e^ Private market uptake of approximately 30% annually. * Annual PCV uptake estimates provided by the surveillance site for the primary series of PCV by 12 months of age (if available, for some sites up to 15 months of age), excluding year of vaccine rollout. ** Annual PCV uptake estimates provided by the surveillance site for the primary series plus the booster dose by 23 months of age, excluding year of vaccine rollout. *** WHO and UNICEF Estimates of National Immunization Coverage (WUENIC) PCV3 uptake, excluding the year of vaccine rollout (PCV3 represents the third dose whether given before 12 months or at or after 12 months, but in some cases uptake estimates may reflect the percentage of surviving infants who received two doses of PCV prior to the first birthday).

**Table 2 microorganisms-09-00696-t002:** Serotype 1 invasive pneumococcal disease all-site weighted average incidence rate ratios comparing the annual post-PCV10/13 incidence rate to the average pre-PCV10/13 incidence rate by age group.

	Year Post-PCV10/13 Introduction									
	0 *	1	2	3	4	5	6	7	8	9
Children <5 Years										
Nnumber of Sites ^a^	37	37	36	36	35	34	33	27	10	3
IRR (95% CI)	1.09 (0.92–1.29)	0.57 (0.48–0.67)	0.29 (0.25–0.35)	0.15 (0.13–0.18)	0.08 (0.07–0.09)	0.05 (0.04–0.05)	0.03 (0.02–0.03)	0.02 (0.02–0.02)	0.01 (0.01–0.02)	0.01 (0.01–0.01)
Children 5–17 Years										
Number of Sites ^a^	34	34	33	33	32	31	30	24	9	2
IRR (95% CI)	1.06 (0.88–1.28)	0.67 (0.55–0.80)	0.41 (0.34–0.49)	0.24 (0.20–0.29)	0.14 (0.11–0.16)	0.08 (0.06–0.09)	0.04 (0.04–0.05)	0.03 (0.02–0.03)	0.01 (0.01–0.02)	0.01 (0.01–0.01)
Adults 18–49 Years										
Numbers of Sites ^a^	29	29	29	29	28	28	27	22	9	2
IRR (95% CI)	0.94 (0.73–1.22)	0.57 (0.44–0.74)	0.34 0.26–0.44)	0.20 (0.15–0.25)	0.11 (0.09–0.14)	0.06 (0.05–0.08)	0.03 (0.03–0.04)	0.02 (0.01–0.02)	0.01 (0.01–0.01)	0.01 (0.00–0.01)
Adults 50–64 Years										
Number of Sites ^a^	29	29	29	29	27	27	27	22	9	2
IRR (95% CI)	0.85 (0.70–1.04)	0.54 (0.44–0.65)	0.33 (0.27–0.40)	0.19 (0.15–0.23)	0.10 (0.08–0.12)	0.06 (0.05–0.07)	0.03 (0.03–0.04)	0.02 (0.02–0.02)	0.01 (0.01–0.01)	0.01 (0.01–0.01)
Adults ≥65 Years										
Number of Sites ^a^	28	28	28	28	27	27	27	22	9	2
IRR (95% CI)	0.82 (0.68–0.99)	0.56 (0.46–0.67)	0.36 (0.30–0.43)	0.20 (0.17–0.24)	0.10 (0.08–0.12)	0.05 (0.04–0.06)	0.03 (0.02–0.03)	0.02 (0.01–0.02)	0.01 (0.01–0.01)	0.01 (0.00–0.01)
All ages										
Number of Sites ^a^	39	39	38	38	37	36	35	29	11	3
IRR (95% CI)	0.98 (0.79–1.21)	0.57 (0.47–0.71)	0.33 (0.27–0.40)	0.18 (0.15–0.22)	0.10 (0.08–0.12)	0.05 (0.04–0.06)	0.03 (0.02–0.04)	0.02 (0.01–0.02)	0.01 (0.01–0.01)	0.01 (0.01–0.01)

PCV: Pneumococcal conjugate vaccine. * Year of PCV10/13 introduction. ^a^ Number of sites with both pre- and post-PCV10/13 data in each post-PCV10/13 year. All-site weighted average IRRs estimated by post-PCV10/13 year and age group using linear mixed-effects regression.

## Data Availability

Restrictions apply to the availability of these data. Data were obtained under data sharing agreements from contributing surveillance sites and can only be shared by contributing organizations with their permission.

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
