# Peer review of "Changes in Invasive Pneumococcal Disease Caused by Streptococcus pneumoniae Serotype 1 following Introduction of PCV10 and PCV13: Findings from the PSERENADE Project"

_microorganisms, 2021, doi:10.3390/microorganisms9040696_

Round 1
Reviewer 1 Report
In this study, authors used data from 52 surveillance sites to demonstrate that after implementation of PCV10 or PCV13 there have been large and sustained decreases in IPD caused by serotype 1 strains among both children targeted for immunization but, through indirect effects, also across unvaccinated children and adults. The manuscript is well written and there is a very good flow between sections, and paragraphs of the text.
There are only two points for authors to address:
It is not clear what was the definition of serotype 1 IPD. Lines 173-174 in Materials and Methods provide information exclusively on how (any serotype) IPD cases were defined in the study.
Authors should consider applying a certain threshold of IPD number for data from a site to be included in the analysis. According to Table 1, there are no pre-PCV10/13 years of data for Latvia (overall, n=5 of ST1 IPD cases post-PCV10/13 implementation), Hong Kong (a single case of ST IPD), and for Italy. Massachusetts, Quebeck, and Hong Kong reported altogether on a single ST1 IPD case each. Since authors do not report on proportions of ST1 among all IPD cases per site, period or year, some of these sites may have extremely low ST1 incidence already prior to PCVs implementation.
Author Response
Reviewer #1
- In this study, authors used data from 52 surveillance sites to demonstrate that after implementation of PCV10 or PCV13 there have been large and sustained decreases in IPD caused by serotype 1 strains among both children targeted for immunization but, through indirect effects, also across unvaccinated children and adults. The manuscript is well written and there is a very good flow between sections, and paragraphs of the text.
We thank the reviewer for their insightful comments which we believe have improved the quality of the manuscript. We have tried our best to improve the manuscript by addressing the reviewer’s comments.
- It is not clear what was the definition of serotype 1 IPD. Lines 173-174 in Materials and Methods provide information exclusively on how (any serotype) IPD cases were defined in the study.
We have clarified the definition of serotype 1 IPD in lines 171-174 of the Methods section.
- Authors should consider applying a certain threshold of IPD number for data from a site to be included in the analysis. According to Table 1, there are no pre-PCV10/13 years of data for Latvia (overall, n=5 of ST1 IPD cases post-PCV10/13 implementation), Hong Kong (a single case of ST IPD), and for Italy. Massachusetts, Quebec, and Hong Kong reported altogether on a single ST1 IPD case each. Since authors do not report on proportions of ST1 among all IPD cases per site, period, or year, some of these sites may have extremely low ST1 incidence already prior to PCVs implementation.
We agree and thank the reviewer for raising this point. We acknowledge that some of the included sites had a very low burden of ST1 IPD, including prior to PCV10/13 introduction. Our statistical methods allow for even a single case from a site to be included and accounts for differences in sample sizes across sites such that the contribution of very small sites to the results is small. The model is driven more by sites with many cases and years of data than sites with few cases or years of data. This is mentioned in lines 227-229 and 254-257 of the Methods section. Excluding sites with very small sample sizes does not meaningfully change the results. We have added this to the results in line 351-352. Because no differences were observed, it was decided by site investigators contributing data and a Technical Advisory Group that it was appropriate to include these sites.
These results were presented to WHO SAGE by pre-PCV10/13 ST1 incidence rates. We generally observed that where the pre-PCV10/13 ST1 burden was low, the impact was greater in the first few years of PCV10/13 roll-out compared to settings where the pre-PCV10/13 burden was high. However, there is no long-term impact (see Figure 3). It was not appropriate to show incidence rate data in the manuscript for two reasons. First, this would reduce anonymization of the site-level data as agreed to in data sharing agreements for the project. Second, surveillance system sensitivity varies across sites and in some sites the incidence rate of the raw data misrepresents the true disease burden. This is especially true in settings where surveillance is challenging to conduct and where the disease burden is highest.
Reviewer 2 Report
The submitted manuscript uses multisite data from multiple sources to determine the changes in serotype 1 distribution globally after the introduction of a PCV into various geographic areas. Data obtained was carefully curated and IPD rates of serotyped data was used to calculate incidence rate ratios for pre and post vaccination timepoints globally. Through this study it was determined that despite high pathogenicity of ST1 strains the us of PCV significantly reduced the rate of IPD cause by this serotype and that this trend was observed in all geographical locations tested. Overall this article is extremely well written and will make substantial contributions to the field and emphasize the effectiveness of PCV introduction. There are a few minor changes that should be correct before publication.
In the discussion there is a good section about limitations and caveats for the data interpretation and the availability of pre-vaccination data from worldwide sources. It would be nice to see some addition to this section about how data analysis of other serotypes may vary or stay the same from the current info presented. ST1 is unique in that it has extremely low carriage rates and is almost always isolated from IPD which makes vaccination efforts particularly effective against this serotype as its spread is limited by reduced nasopharyngeal carriage. Extrapolation to a larger setting and what these findings mean in context of total IPD and not just ST1 will be a good addition to the current submission.
Ln 157. “case” instead of “cases”
Ln 306. “PCV13” not “PCV3”
Author Response
Reviewer #2
- The submitted manuscript uses multisite data from multiple sources to determine the changes in serotype 1 distribution globally after the introduction of a PCV into various geographic areas. Data obtained was carefully curated and IPD rates of serotyped data was used to calculate incidence rate ratios for pre and post vaccination timepoints globally. Through this study it was determined that despite high pathogenicity of ST1 strains the use of PCV significantly reduced the rate of IPD cause by this serotype and that this trend was observed in all geographical locations tested. Overall, this article is extremely well written and will make substantial contributions to the field and emphasize the effectiveness of PCV introduction. There are a few minor changes that should be correct before publication.
We thank the reviewer for pointing out that this study is of interest to the field and that the analyses were undertaken with care. We have tried our best to answer the items raised in order to improve the manuscript.
- In the discussion there is a good section about limitations and caveats for the data interpretation and the availability of pre-vaccination data from worldwide sources. It would be nice to see some addition to this section about how data analysis of other serotypes may vary or stay the same from the current info presented. ST1 is unique in that it has extremely low carriage rates and is almost always isolated from IPD which makes vaccination efforts particularly effective against this serotype as its spread is limited by reduced nasopharyngeal carriage. Extrapolation to a larger setting and what these findings mean in context of total IPD and not just ST1 will be a good addition to the current submission.
We agree with this important point and currently have several additional manuscripts in progress which will evaluate the impact on all-serotype, vaccine-type, and non-vaccine-type IPD using the same dataset. This has been added to the Discussion section in lines 477-481.
- Ln 157. “case” instead of “cases”
This has been corrected.
- Ln 306. “PCV13” not “PCV3”
This is correct as, “PCV3”. This is in reference to the WUENIC PCV3 vaccine coverage estimate definition. Here, PCV3 refers to the third dose of either PCV10 or PCV13 whether given before or at or after 12 months of age.